# Highly efficient genome editing by CRISPR-Cpf1 using CRISPR RNA with a uridinylate-rich 3′-overhang

Su Bin Moon[1,2], Jeong Mi Lee[1,2], Jeong Gu Kang[1], Nan-Ee Lee[1,2], Dae-In Ha[1,2], Do Yon Kim[1,2], Sun Hee Kim[1], Kwangsun Yoo[1,2], Daesik Kim[3], Jeong-Heon Ko[1,2] & Yong-Sam Kim[1,2]

Genome editing has been harnessed through the development of CRISPR system, and the CRISPR from *Prevotella* and *Francisella* 1 (Cpf1) system has emerged as a promising alternative to CRISPR-Cas9 for use in various circumstances. Despite the inherent multiple advantages of Cpf1 over Cas9, the adoption of Cpf1 has been unsatisfactory because of target-dependent insufficient indel efficiencies. Here, we report an engineered CRISPR RNA (crRNA) for highly efficient genome editing by Cpf1, which includes a 20-base target-complementary sequence and a uridinylate-rich 3′-overhang. When the crRNA is transcriptionally produced, crRNA with a 20-base target-complementary sequence plus a $U_4AU_4$ 3′-overhang is the optimal configuration. U-rich crRNA also maximizes the utility of the AsCpf1 mutants and multiplexing genome editing using mRNA as the source of multiple crRNAs. Furthermore, U-rich crRNA enables a highly safe and specific genome editing using Cpf1 in human cells, contributing to the enhancement of a genome-editing toolbox.

[1] Genome Editing Research Center, KRIBB, 125 Gwahak-ro, Daejeon 34141, Republic of Korea. [2] Department of Biomolecular Science, KRIBB School of Bioscience, Korea University of Science and Technology (UST), 217 Gajeong-ro, Yuseong-gu, Daejeon 34113, Republic of Korea. [3] Department of Chemistry, Seoul National University, Seoul 08826, Republic of Korea. These authors contributed equally: Su Bin Moon, Jeong Mi Lee. Correspondence and requests for materials should be addressed to Y.-S.K. (email: omsys1@kribb.re.kr)

Genome engineering has been surprisingly harnessed through the development of the CRISPR/CRISPR-associated protein (Cas) system[1,2], which serves as a form of bacterial immunity against viral invasion[3,4]. Targeted gene knockout and knock-in have been demonstrated in a variety of species and cell types[5,6]. In particular, one-step generation of animals with targeted gene modifications has recently been implemented within half a year, which previously required more than 1 year until the establishment of a founder[7]. Several efforts have been made to develop novel plants or crops with improved traits by editing endogenous genes, which are expected to overcome the negative images of conventional gene-modified organisms[8,9]. Moreover, the CRISPR system is also expected to provide a new therapeutic modality for various indications, and pre-clinical and clinical trials are increasingly underway worldwide[10,11]. Considering these diverse applications and biological systems, the development of varied genome-editing tools needs to provide an arsenal of the most appropriate tools in the genome-editing toolbox.

The genome-editing toolbox has recently been diversified by newly identified CRISPR systems with a single effector nuclease, including CRISPR from *Prevotella* and *Francisella* 1 (Cpf1), C2c1, C2c2, and C2c3 effector nucleases[12–15]. CRISPR-Cpf1 is an RNA-guided, class II CRISPR/Cas system that is analogous to CRISPR-Cas9, but shows unique features distinct from those of the CRISPR-Cas9 system[12]. Despite the presence of several Cas9 orthologs with a smaller gene size such SaCas9[16] and CjCas9[17], Cpf1 is generally smaller than most of Cas9 orthologs that has a protospacer adjacent motif (PAM) sequence with a high frequency in the genome, which is favorable particularly for clinical purposes. Moreover, Cpf1 relies on a T-rich PAM sequences at the 5′-end of the protospacer sequence, in contrast to the G-rich sequences for Cas9. Recently, the range of targetable genes by Cpf1 was expanded by engineered Cpf1 variants[18]. In addition, Cpf1 creates double-strand breaks in a staggered manner at the PAM-distal position, which may provide additional advantages particularly for knock-in strategies. The lower off-target incidence compared with that of CRISPR-Cas9 is thought to provide additional advantages regarding safety issues[19–21]. Despite these multi-dimensional merits, the adoption of CRISPR-Cpf1 has not been as explosive as expected for the past 2 years. We assume that this lower-than-expected adoption rate, at least partly, stemmed from an overall lower and highly deviated indel efficiency compared with CRISPR/Cas9. Thus, if this sole shortcoming is improved, then CRISPR-Cpf1 would become a more versatile tool that can be widely used in various experimental and clinical settings.

Engineering of guide RNA (gRNA) has been performed to enhance or diversify the functions of programmable nucleases together with the engineering of effector proteins themselves. Truncated gRNAs have improved the specificity of Cas9[22], and aptameric extensions of gRNA have been proposed to confer the function of gene-targeted transcriptional regulations to the CRISPR system[23]. gRNA for Cas9 was also optimized by extending the duplex length and replacing the fourth thymine with adenine or guanine, which improved the knockout efficiency in cells[24]. For Cpf1, the indel activity tolerated several chemical modifications of the CRISPR RNA (crRNA) at the 3′-termini and 5′-termini[25], but the modification itself did not significantly improve the efficiency of AsCpf1 in vivo. Here, we report an enhanced CRISPR/Cpf1 system, in which crRNA has a repeat sequence and a 20-nt target-complementary sequence plus a uridinylate-rich 3′-overhang. For synthetic crRNA, the addition of 8-meric uridinylates ($U_8$) created the most efficient indel efficiency. For transcribed crRNA, $T_4AT_6$ created a maximum indel efficiency of Cpf1 when added as a 3′-overhang after a 20-

base target sequence in the template DNA. The engineered U-rich crRNA enabled a highly efficient and specific genome editing by Cpf1. This engineered CRISPR-Cpf1 system will significantly contribute to enhancing the genome-editing toolbox.

## Results

**A U-tail of crRNA enhances AsCpf1 activity in vitro.** Structural analysis of the crRNA-Cpf1 complex and target DNA was performed to clarify how Cpf1 induces double-strand (ds) DNA breakage guided by crRNA at targets with T-rich PAM sequences[26,27]. However, Dong et al.[26] and Yamano et al[27]. have invariably reported that 3–4 nucleotide (nt) residues of crRNA and the target DNA remain unidentified possibly due to a high flexibility. This may suggest that the critical nucleotide length in crRNA, which is necessary for target recognition and specific binding, would be ca. 20-nt consistent with that for CRISPR-Cas9. Thus, we tested whether the 3′-end 3–4 nts in crRNA are essential components. For this, we transfected a plasmid vector with a codon-humanized *AsCpf1* gene and crRNA-expressing PCR amplicons into human embryonic kidney-293T (HEK-293T) cells. crRNA was designed to comprise a 20-nt target sequence for the *DNMT1* gene followed by three variable sequences. For an initial check, we tested four different crRNAs, each of which contained a 3′-overhang of AAA ($A_3$), $U_3$, $G_3$, or $C_3$ as variables. Interestingly, crRNA with a $U_3$ 3′-overhang showed the highest indel efficiency (Fig. 1a). Moreover, this crRNA configuration led to an improved indel efficiency compared with the crRNA carrying a 23-nt target-complementary sequence. This result was identically obtained from experiments conducted for three additional target genes (Supplementary Figure 1). An in vitro DNA digestion assay revealed that the crRNA carrying the $U_3$ 3′-overhang resulted in a significantly improved double-stranded DNA (dsDNA) breakage compared with the G-rich one (Fig. 1b). This result further prompted the pursuit of the optimal configuration of crRNA in an unbiased manner. For this, we prepared the plasmid DNA library that encodes crRNAs with a library of 3′-overhang. crRNA library oligonucleotides with a 11-nt 3′-end sequence library ($4^{11}$) were synthesized and quality controlled so that each crRNA occupies the equal molar ratio. Each crRNA was designed to have a 17-nt on-target sequence and an 11-nt (N11) randomized nucleotide sequence (Supplementary Figure 2). Such design was aimed to define the essential on-target length and additional regulatory sequence. The negative selection method[28] was adopted to pursue the optimal configuration of crRNA, in which *Escherichia coli* cells carrying an efficient crRNA are less prone to survive in the ampicillin-supplemented agar plates. The survived E. coli cells were collected to extract crRNA-encoding plasmid DNA, and the deep sequencing analysis was performed to calculate the counts of nucleotides at each position in the target region (Fig. 1c). The deep sequencing data analysis revealed that the crRNA-encoding plasmid DNA library was made so that A, T, G, and C occupied almost equal molar ratio at each position as assessed by dCpf1 treatment. Marginal variations were normalized by the values obtained by dCpf1 treatment. In contrast, there were significant differences in the frequency of each nucleotide in a position-dependent manner when AsCpf1 was treated. The probability values were derived from the inverted values of the nucleotides ratios at each position, which indicates the configuration of the optimal crRNA (Fig. 1d). The result indicated that a 20-nt on-target sequence is critical, but should be followed by a uridine-rich 3′-tail independently of the on-target sequence from the position of 21. To gauge the effective length of the uridine tail, we used a chemically synthesized crRNA with different lengths of 3′-proximal uridinylate and tested the in vitro dsDNA breakage efficiency of the AsCpf1-

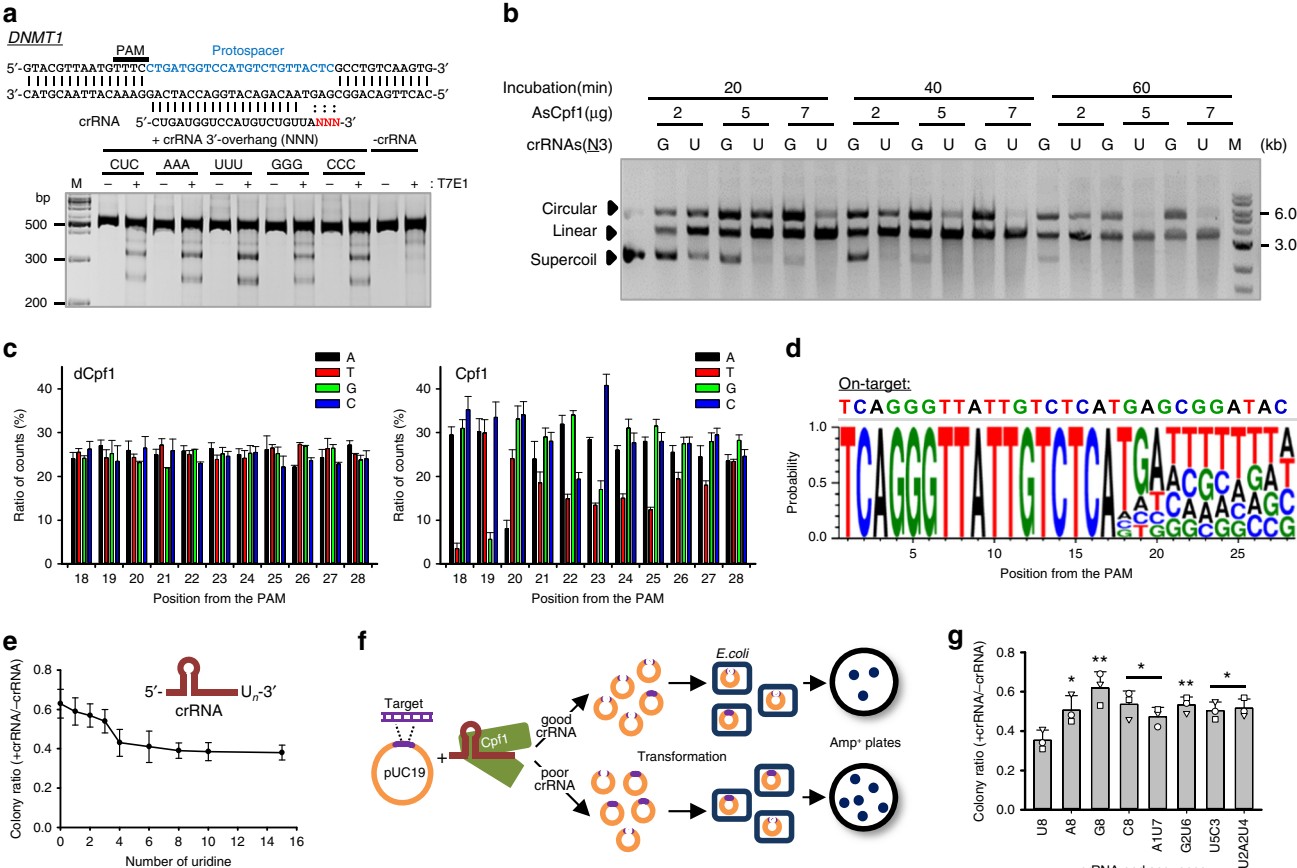

**Fig. 1** Improved dsDNA breakage efficiency of Cpf1 in vitro by U-rich crRNA. **a** Dependence of the indel efficiency of AsCpf1 on the 3′-overhang sequence of crRNA. Variations of the 3′-terminal three bases of crRNA affected the AsCpf1 indel activity in vivo, and the presence of 3-mers of uridine was the most critical for the improved AsCpf1 activity. **b** Improved in vitro dsDNA cleavage activity of AsCpf1 by the 3′-end uridine-rich crRNA. Vector constructs (500 ng) were digested in vitro in the presence of a ribonucleoprotein complex of AsCpf1 and crRNA (50 ng) at various incubation times (20–60 min) with variable amounts of AsCpf1 (2–7 μg) in a 20-μL reaction. **c** Positional ratio of nucleotide counts of crRNA template DNA in the protospacer region as assessed by deep sequencing analysis. The negatively selected E. coli cells carry less-efficient crRNA-encoding plasmid DNA, thereby rendering higher count numbers in deep sequencing. dCpf1 was used to normalize the variations in the crRNA library synthesis and experimental procedures ($n = 3$). All data are presented as means±standard deviations. **d** Optimal configuration of AsCpf1-crRNA for yielding the highest indel efficiency. Twenty-nt target sequence should be followed by uridine-rich sequences independently of the subsequent on-target sequences in the crRNA for an optimal AsCpf1 indel activity. **e** Dependence of the length of the 3′-end uridines of the crRNA on the in vitro AsCpf1 activity. The increment of the length of uridines of the chemically synthesized crRNAs led to the improved AsCpf1 activity in vitro with up to eight bases. **f** Scheme for an in vitro dsDNA breakage activity assay. Partially digested plasmids were used to transform DH-5α E. coli cells, and the AsCpf1 activity was calculated from the number of colonies formed on ampicillin-containing plates. **g** Validation of the highly efficient activity of AsCpf1 by the U-rich 3′-overhang in crRNA. Replacements of the uridines with non-uridine bases at any location and at any number resulted in decreased AsCpf1 activity. Data are presented as the mean±standard deviation. $*p < 0.05$, $**p < 0.01$, compared with U8 ($n = 3$), two-tailed Student's t test

crRNA ribonucleoproteins. The maximal DNA cleavage was obtained for crRNA with a U8 overhang (Fig. 1e). The further increments of the uridine length did not significantly affect the dsDNA cleavage. Thus, we concluded that the addition of eight uridinylates to a 20-nt target-complementary sequence had the optimal dsDNA cleavage efficiency in vitro.

This result was validated by designing an in vitro experiment in which a pUC19 plasmid vector with a 23-nt target sequence for *DNMT1* was incubated with an equimolar ratio of the AsCpf1-crRNA ribonucleoprotein. After a partial digestion for 1 h, the digested plasmid vector was used to transform E. coli DH-5α. The E. coli cells were plated onto ampicillin-containing Lysogeny Broth (LB) agar plates, and the number of colonies formed was counted (Fig. 1f). Repetitive experiments revealed that the addition of eight uridinylates was critical for a highly efficient AsCpf1 activity (Fig. 1g). Replacing uridine with any other nucleotide at any location decreased the dsDNA cleavage activity

of AsCpf1. A robust assay that we used with modifications to achieve an optimal configuration of the crRNA[28] also confirmed the effectiveness of the U-rich crRNA (Supplementary Figure 3). Taken together, these results suggest that the 3′-end U-rich tail in crRNA is a critical structural determinant for highly efficient dsDNA breakage by AsCpf1.

**crRNA with the U-tail improves Cpf1 activity in vivo.** Next, we tested whether the U-rich crRNA is applicable for highly efficient genome editing in vivo. This question was addressed by investigating the indel efficiency in HEK-293T cells into which a vector construct carrying the codon-humanized *AsCpf1* gene was co-transfected with crRNA-encoding PCR amplicons consisting of the U6 promoter, 20-nt target-complementary sequence and 3′-end variable sequences (Fig. 2a). The DNMT1 gene was targeted with the crRNA-encoding PCR amplicons carrying a 20-nt

matched target sequence plus 4 additional 3′-overhangs ($A_4$, $G_4$, $T_4$, $C_4$, or 4 additional target-matched nucleotides). Consistent with the in vitro result (Fig. 1), the use of the 3′-end U-rich crRNA led to a significantly improved indel efficiency in vivo compared with crRNAs with 20-nt on-target ($20_t$), $20_t$ plus $A_4$, $G_4$, or $C_4$, or even crRNA with an extended length ($24_t$) of on-target sequence (Fig. 2b). It could be assumed that the presence of multiple residues of uridinylates confers stability to the crRNA inside cells, and that the enhanced stability of crRNA is responsible for the increased indel efficiency of AsCpf1. However, the presence of $T_6$ at the 3′-end of a single-guide RNA-encoding PCR amplicon did not affect the indel efficiency of the CRISPR-Cas9 system (Fig. 2c). In addition, because the U-rich 3′-overhang was effective in the in vitro system (Fig. 1), we concluded that the U-rich overhang may regulate the Cpf1 activity when crRNA binds to Cpf1.

Next, we investigated the dependence of AsCpf1 activity in vivo on the length of the U-tail. The increased length of uridinylates, up to an 8 mer, proportionally improved Cpf1 activity (Fig. 1). In contrast to the in vitro data, the indel efficiency was almost saturated when the additional length of T was 4. Further increments of the length of T in the crRNA-encoding PCR amplicon did not affect the indel efficiency (Fig. 2d). In fact, this result is explained by the fact that RNA polymerase III governs U6-promoted gene transcription. In this process, a consecutive T-rich sequence ($T_5$ or $T_6$) in a template DNA acts as a termination signal, and as a result, four uridinylates ($U_4$) are generated at the 3′-end[29]. Thus, further increments of the length of the thymidine base sequence in the template do not lead to concomitant increments of the uridine length in the crRNA. However, when using chemically synthesized crRNAs, an increased length of the uridinylates of up to 8-mers was proportionally associated with

the improved Cpf1 activity as observed in the in vitro experiment (Fig. 1c). This contradictory result led us to further engineer crRNA for application to in vivo genome engineering. Considering that the enriched uridinylate 3′-overhang in crRNA was critical for improved efficiency, we designed a crRNA-encoding template DNA so that 4 deoxythymidinylates ($T_4$) are followed by one non-T base plus $T_6$, generating a crRNA carrying a $U_4VU_4$ 3′-tail, where V is A, C, or G. A $U_4$-tail is indeed produced in the transcripts from the T-rich termination sequences ($T_5$ or $T_6$) of templates[29]. As expected, the incorporation of A into Ts led to higher indel efficiency compared with that of G and C (Fig. 2e). A further increase in the number of U was attempted by adding more U4A units; however, this addition did not further increase the efficiency. From these results, we concluded that for synthetic crRNA, the addition of at least eight uridinylates ($U_8$) after a target-complementary sequence is critical for highly efficient genome editing. When crRNA is transcriptionally produced from a DNA template, the template sequence should carry a "TTTTATTTTTT" sequence after a target-matched sequence. This configuration generates a $U_4AU_4$ 3′-overhang in the crRNA, which was almost comparable to the synthetic $U_8$-crRNA in indel efficiency. The most efficient target length was 20 (±1) nt, depending on the target (Fig. 2f). This optimized crRNA configuration was identically applied to Cpf1 from *Lachnospiraceae bacterium* (LbCpf1) (Fig. 2g)[12]. The importance of improved Cpf1 activity by the U-rich crRNA was clearly seen in a "knock-in" experiment. The overall knock-in efficiency for the CRISPR-Cpf1 system was lower than that for CRISPR-Cas9, even when a single-stranded oligonucleotide (ssODN) was used as a donor[30], and only U-rich crRNA enabled a detectable level of the ssODN-based knock-in by AsCpf1 (Supplementary Figure 4).

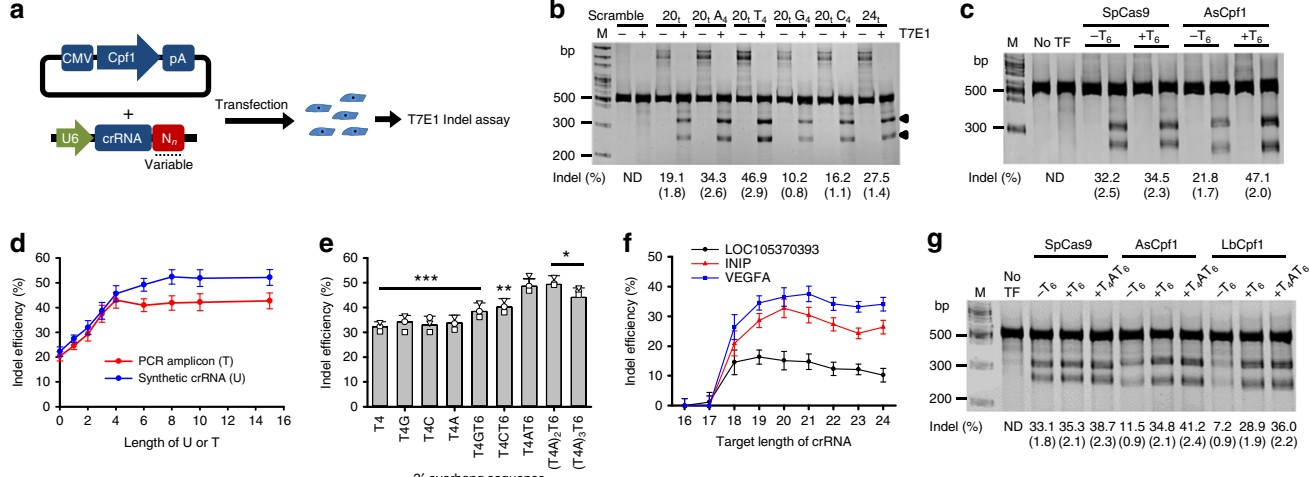

**Fig. 2** Optimized configuration of crRNA for highly efficient genome editing in vivo. **a** Scheme of the in vivo assay to determine the most efficient configuration of the crRNA. Cpf1-encoding plasmids were co-transfected with crRNA-encoding PCR amplicons into HEK-293T cells, and the Cpf1 activity was assessed by T7E1 indel assays. **b** Improved indel efficiency of AsCpf1 in vivo by the U-rich 3′-overhang following a 20-nt target-complementary sequence in transcribed crRNAs. This gel image is a representative result of three repeated experiments. Indel values are mean±standard deviation. **c** Improved indel efficiency by the 3′-end U-rich guide RNA as a unique feature of Cpf1. 3′-Proximal addition of uridinylates did not change the indel efficiency of SpCas9 in vivo. This gel image is representative of three repeated experiments. Indel values are the mean±standard deviation. **d** Improved indel efficiency of AsCpf1 in vivo by increased uridinylate lengths. The AsCpf1 activity was improved by the increased lengths of 3′-end uridinylates up to 8–10 mers for the chemically synthesized crRNA and up to six bases for the crRNA-encoding PCR amplicons. **e** Optimized 3′-end configuration of crRNA for highly efficient genome editing. Addition of the U4AU4 3′-overhang in crRNA maximized the indel efficiency of AsCpf1. *$p > 0.05$, **$p < 0.05$, ***$p < 0.01$ ($n = 3$), two-tailed Student's $t$ test. **f** The optimal target length for the use of a U-rich crRNA. A target length of 20 (±1) nt was optimal for the U-rich crRNA. **g** Validation of the optimal crRNA configuration for highly efficient genome editing using CRISPR-Cpf1. The optimal configuration of the $U_4AU_4$ 3′-overhang in addition to a 20-nt target-matched sequence was identically applied for LbCpf1 as well as AsCpf1, but not for SpCas9. This gel image is representative of three repeated experiments. Indel values are the mean±standard deviation

**Large-scale investigation of Cpf1 and Cas9 efficiency**. There is a concern that the improved indel efficiency by the U-rich crRNA is sequence dependent, and not generally applicable to a wide range of targets. To address this issue, we attempted to investigate the indel efficiency of AsCpf1 at a large scale in a biased manner and to compare the indel efficiency with that obtained by SpCas9. To rule out differences reflecting the target-dependent indel efficiency, we searched for targets that are common for Cpf1 and Cas9 with the query sequence of 5′-TTTV(N)$_{20}$NGG-3′. Such targets have PAM sequences for both AsCpf1 and SpCas9 and share 20-nt target sequences. We randomly selected 115 targets that were PCR validated in HEK-293T cells, comprising 49 exons, 32 introns, and 34 intergenes. The target information is provided in Supplementary Table 1. The single-guide RNAs (sgRNAs) and crRNAs were designed to be transcribed from PCR amplicons encompassing the U6 promoter and sgRNA or crRNA sequences carrying the respective target sequences.

Figure 3a, b shows the dot and box-and-whisker plots, respectively, for the indel efficiencies at the investigated targets. For each target, the indel efficiencies of SpCas9, AsCpf1 with canonical crRNA (Con-AsCpf1), and AsCpf1 with U-rich crRNA (U_rich-AsCpf1) were investigated. Two out 115 targets had no indel mutations by any of the gene-editing systems, but the remaining 113 targets showed detectable levels of indel mutations by at least one of the tested systems (98.2% coverage). To our knowledge, we could, for the first time, investigate a sufficient sample size with statistical power and compare the indel efficiencies of Cas9 and Cpf1. Statistical analysis of these large-scale data led to the following conclusions: (1) The overall efficiency of AsCpf1 guided by canonical crRNA was lower than that of SpCas9 ($p = 0.003$), despite the previous report that the efficiency of Cpf1 is comparable to that of SpCas9[17]. (2) U-rich crRNA contributed to significantly enhancing the indel efficiency of AsCpf1 ($p = 0.00003$), and the AsCpf1 efficiency enhanced by U-rich crRNA was almost comparable to the SpCas9 efficiency ($p = 0.29$). (3) For targets with detectable mutations, 90.3% (94/104) of the targets experienced an increased efficiency by the U-rich crRNA with the increments ranging from 1.07-fold to 12.98-fold and an average fold of 2.31. (4) Cpf1 and Cas9 were complementary to each other as genome-editing tools. AsCpf1 guided by the U-rich crRNA efficiently resulted in mutations for targets with no or a low indel efficiency by SpCas9, and vice versa. (5) The stimulatory effects of the U-rich crRNA on the Cpf1 activity was more pronounced for targets with low indel efficiency than those with high efficiency. For targets with the innately high efficiency, the U-rich effect was marginal. These results suggest that the CRISPR-Cpf1 system, which uses U-rich crRNA in a highly efficient and predictable manner, could be used as a complementary genome-editing tool to the CRISPR-Cas9 system.

**No compromise in off-target effects by the U-rich crRNA**. The high levels of target specificity and low off-target activity of Cpf1 have been reported in several studies through several independent off-target assessment methods[19–21]. These reports have inarguably claimed that both AsCpf1 and LbCpf1 are highly specific for genome editing in human cells and show a lower off-target activity than SpCas9. Because the shortened on-target length (from 23 to 20 nt) may compromise the high target specificity of Cpf1, thereby resulting in an increased off-target activity. we compared the off-target activity of AsCpf1 guided by the U-rich crRNA with that guided by the canonical 23-nt target-complementary crRNA by employing both a biased and an unbiased approach.

Using Cas-OFFinder[31], we selected nine potential off-target sites with the smallest bulge and mismatch against an on-target

sequence of phospho-tyrosine kinase 6 (*PTK6*). Subsequently, any incidence of mutations by AsCpf1 was investigated for those potential off-targets in HEK-293T cells, and the indel efficiency difference was compared between the canonical 23-nt crRNA (con-crRNA) and our U-rich crRNA. Consistent with the results shown in Figs. 2 and 3, deep sequencing analysis revealed that the indel efficiency in this on-target site was increased by 2.19-fold by the U-rich crRNA (Fig. 4a). However, we did not observe any indel mutation inside the target sequences for all potential off-targets. Any aberration to the reference sequences is likely a product of single-nucleotide polymorphism (SNP) because the SNP was identically observed and at similar levels in AsCpf1 non-treated cells (Supplementary Table 2). These results suggest that the use of the U-rich crRNA did not affect the off-target activity of AsCpf1. Next, we investigated the levels of off-target at a *DNMT1* site resulting from the use of crRNAs with single base mismatches against the protospacer sequence. Significant and considerable levels of tolerance were previously observed for mismatches at the 3′-end and the middle regions (positions 8–10) of the crRNA, respectively,[19]. Repeated experiments revealed the widespread occurrence of off-target indel mutations throughout the target positions, although the aforementioned positions showed higher off-target levels (Fig. 4b). Interestingly, the use of the U-rich crRNA effectively lowered the tolerance of single base mismatches for most of the target positions, except for the 3′-end region (thus, positions 18–20). This result is consistent with a previous report suggesting that a truncated gRNA is responsible for the improved target specificity of SpCas9[20]. We observed significantly higher levels of the off-target activity at positions 18–20, and the U-rich crRNA slightly exacerbated the off-target activity at these regions. Nonetheless, the use of a U-rich crRNA did not significantly compromise the inherent level of the Cpf1 specificity.

Finally, we performed an unbiased, genome-wide analysis of the Cpf1 specificity to monitor any change in the off-target activity according to the crRNA structures using Digenome-seq analysis[20]. Cell-free genomic DNA isolated from HEK-293T cells was subjected to in vitro cleavage by the AsCpf1-crRNA ribonucleoprotein complex. The quantitative real-time PCR analysis revealed that more than 98% of the genomic DNA was digested by the AsCpf1-U-rich crRNA as well as by the AsCpf1 canonical crRNA ribonucleoprotein complex (Supplementary Figure 5). The cleaved products were then subjected to whole genome sequencing and the sequence data were aligned against the human reference genome database (GRCh38.p11). Integrative genomic viewer confirmed the typical cleavage pattern at positions 18–20 of the non-target strand and 22 of the target strand. Computer analysis was performed to find off-target sites using the Digenome-seq program, which were identified with DNA cleavage score ≥2.5 and with six or fewer mismatches. The identified sites are listed in Supplementary Table 3, and the off-target sites were plotted on the genome-wide Circos plot (Fig. 4c). The number of off-target sites for canonical and U-rich crRNA was not significantly different: 41 and 46 sites were identified for canonical and U-rich crRNA, respectively, and 30 of them were commonly identified (Fig. 4d). The absence of crRNA did not produce any cleavage site with a significant DNA cleavage score (>2.5), confirming the crRNA-dependent DNA cleavage. Moreover, the overall off-target pattern on the genome-wide Circos plot was almost identical for both crRNAs. The sequence logo analysis also shows an identical pattern, in that the PAM-proximal sequences were identically conserved and the tolerance was higher in the PAM-distal sequences for both crRNAs (Fig. 4e). These data suggest that the high specificity of AsCpf1 was not compromised by the U-rich 3′-overhang.

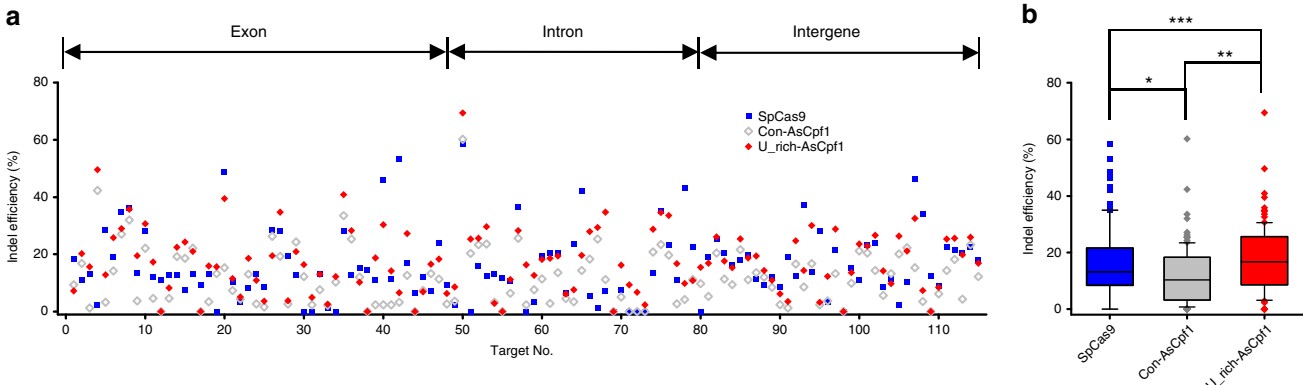

**Fig. 3** Large-scale comparison of the genome-editing efficiencies of CRISPR-AsCpf1 and CRISPR-SpCas9. **a** A dot plot of the indel efficiencies of AsCpf1 and SpCas9 in HEK-293T cells. The indel efficiencies of AsCpf1 and SpCas9 were compared on common targets with a sequence of 5'-TTTV(N)$_{20}$NGG-3', where V is A, C, or G. For the AsCpf1 activity, the conventional crRNA (for con-AsCpf1) configuration was compared with our optimized U-rich crRNA (for Opt-AsCpf1). **b** A box-and-whisker plot for the indel efficiencies of AsCpf1 and SpCas9. The graph consists of median values, lower and upper quartile, and outer standard deviations. *$p = 0.003$, **$p = 0.00003$, ***$p = 0.29$, two-tailed Student's $t$ test

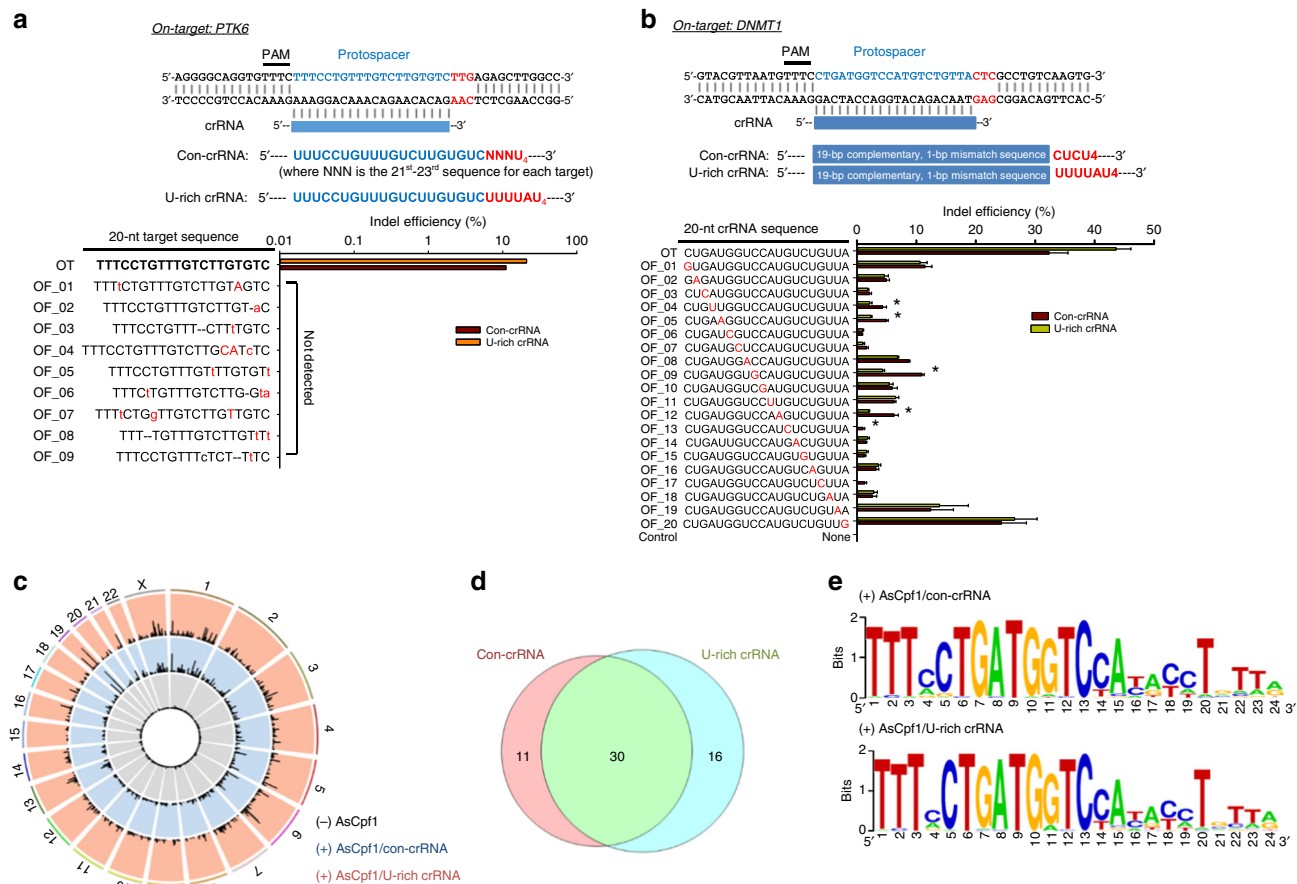

**Fig. 4** No influence of the U-rich crRNA on off-target effects. **a** Comparison of the off-target activity of conventional and U-rich crRNA-guided AsCpf1 at potential off-target sites. Off-target activity was measured by deep sequencing at potential off-target sites with <2 bulges and 2 mismatches against an on-target sequence of phospho-tyrosine kinase 6 (*PTK6*). The use of U-rich crRNA did not harm the specificity of AsCpf1, whereas it contributed to the increased on-target activity of Cpf1. **b** Comparison of the off-target activity of AsCpf1 between the conventional and U-rich crRNA with a one-base mismatch against an on-target sequence. The indel efficiency was calculated from the T7E1 indel assays. The use of a U-rich crRNA effectively lowered the tolerance of a single base mismatch at several positions, thereby contributing to the enhanced AsCpf1 specificity. *$p < 0.05$, compared with U-rich crRNA, two-tailed Student's $t$ test. **c** The genome-wide Circos plot for the off-target sites identified by Digenome-seq analysis with a DNMT1 on-target site. **d** The number of off-target sites for canonical and U-rich crRNA. **e** The sequence logo analysis for recognition of the target sequence. The recognition pattern was equally conserved for the both crRNAs

**Applications of the U-rich crRNA.** Recent outstanding studies have expanded the utility of CRISPR-Cpf1 as an alternative genome-editing tool to CRISPR-Cas9. One of the reports involved the Cpf1-excised generation of crRNAs from mRNA transcripts in mammalian cells[32], which has made multiplexed genome editing easier. We attempted to investigate whether the U-rich crRNA is applicable to this strategy. An array of crRNA sequences, each of which had a 23-nt on-target-complementary sequence, was incorporated into the 3′-untranslated region of the eGFP gene. For comparison, the T-rich components were inserted between a 20-base target and the scaffold of the adjacent crRNA (Fig. 5a). Three of the targets included in the large-scale valida-tion study (Fig. 3) were investigated and found to show indel efficiencies that were almost similar to those that were indivi-dually investigated. More importantly, the incorporation of the U-rich sequence invariably improved the indel efficiencies for the investigated targets. The extent of the efficiency increments was also similar to that of the individual tests.

Another advancement was introduced by Gao et al.,[18] who engineered two AsCpf1 PAM variants carrying S542R/K607R (RR variant) and S542R/K548V/N552R (RVR variant) mutations. The two variants significantly lowered the hurdles of the inherently limited target range of Cpf1. We determined whether the U-rich crRNA could improve the indel efficiency of the two AsCpf1 variants as seen for the wild-type AsCpf1. First, we selected three loci as common targets for WT AsCpf1 and the RR variant with TTTA PAM at one strand and TYCC (two TTCC and one TCCC) at the other strand (Fig. 5b). At all three targets tested, the U-rich crRNA improved the indel efficiency of AsCpf1. At targets 1 and 2, the RR variant showed a higher indel efficiency compared with the WT AsCpf1 when guided by the canonical crRNA but experienced a mitigated improvement when guided by the U-rich crRNA. However, at another target (target 3), the U-rich crRNA showed a more dramatic improvement of the indel efficiency for the RR variant. Next, the RVR AsCpf1 variant was also compared to the WT AsCpf1. The RVR variant retained the traits of TTTV PAM recognition, and thus a single target with a TTTA PAM was shared by WT and RVR variants (Fig. 5c). As expected, the U-rich crRNA improved indel efficiency for both the WT and the RVR variants. Although the percentage improvement was different among the targets in this case, the stimulatory effect of the U-rich crRNA was invariably observed, irrespective of the identity of the targets and AsCpf1 form. Collectively, the U-rich crRNA can be used in a versatile manner for the genome editing of multiple targets and for using Cpf1 variants in mammalian cells, thereby making the CRISPR-Cpf1 system a more attractive genome-editing tool with wide applicability.

**Improved binding affinity of the AsCpf1-U-rich crRNA com-plex.** We attempted to make it clear whether the improved Cpf1 activity may arise primarily from the enhanced stability of the crRNA or from the direct regulation of Cpf1. If the stability of crRNA is mainly responsible for the improved Cpf1 activity, there would be a difference in the pattern or endogenous levels of crRNA upon transfection of PCR amplicons. To test this, crRNA levels were traced by northern blot analysis (Fig. 6a). Repetitive experiments, however, found no significant increase in the endogenous crRNA levels by the U-rich 3′-overhang. Additional-ly, to rule out the involvement of ribonucleases and differential degradation of crRNA according to the 3′-overhang, we used a chemically modified gRNA for both Cas9 and Cpf1, in which the 3′-terminal four nucleotides were covalently linked with the phosphorothioate group. This treatment would basically prevent the degradation of the gRNAs by rioexonucleases[33] and thus

make it possible to investigate the effect of U-rich 3′-overhang by excluding the nuclease tolerance issue. The chemically modified U-rich crRNAs exhibited an even higher Cpf1 activity compared with the chemically modified canonical crRNA. On the other hand, there was still no difference between chemically modified gRNAs for Cas9 (Fig. 6b). In addition, Karvelis et al.[34] reported that the minimal length of tracrRNA for full SpCas9 activity is approximately 63 nt and that the shorter length (for instance, 58 nt) resulted in the mitigated activity. If the poly-uridinylates affect the stability of gRNA in cells, the U-rich 3′-overhang in the shorter tracrRNA would improve SpCas9 activity. However, the presence of U4AU4 in the shorter tracrRNA did not induce any improved Cas9 activity. Rather, the poly-uridinylate down-regu-lated the SpCas9 activity for 63-nt tracrRNA. These data con-vinced us that the major reason for the improved activity by the U-rich 3′-overhang is not the stability effect.

Additionally, we attempted to demonstrate that the U-rich 3′-overhang contributes to the favorable binding of the crRNA to a Cpf1 molecule by adopting two independent methodological approaches. First, the microscale thermophoresis (MST) techni-que was adopted to assess the binding properties of the effector proteins (SpCas9 and AsCpf1) and their gRNAs. MST is based on the directed movement of molecules along a temperature gradient, an effect termed "thermophoresis"[35]. The thermo-phoretic behaviors of a protein typically differ significantly from the thermophoresis of a protein–ligand complex due to binding-induced changes in size, charge, and solvation energy. By measuring the change in the normalized fluorescence ($F_{norm}$) of a ligand (here, Cy5-labeled gRNAs) titrated against their binding effector proteins, a dissociation constant $K_d$ can be derived by plotting $F_{norm}$ against the concentration of the titrants. As shown below, the U-rich 3′-overhang resulted in a significantly improved binding affinity for AsCpf1 compared with the canonical crRNA. However, the U-rich 3′-overhang did not induce any detectable differences in binding properties for the sgRNA-SpCas9 complex (Fig. 6d). To obtain more quantitative insights, we conducted isothermal titration calorimetry (ITC) analysis (Fig. 6e) in which the crRNA was titrated in the presence of AsCpf1. More drastic heat changes were observed by the U-rich crRNA, thereby causing an increased binding constant by 16.2-fold ($K_a = (1.90 \pm 0.87) \times 10^8$ M$^{-1}$ for the U-rich crRNA versus $(1.15 \pm 0.54) \times 10^7$ M$^{-1}$ for the canonical crRNA). The $\Delta H$ was $-31.92 \pm 1.79$ and $-22.86 \pm 1.86$ kcal/mol for the U-rich and canonical crRNA, respectively. Considering the calculated $\Delta S$ was $-69.2$ and $-44.4$ cal/mol/deg for the U-rich and canonical crRNA, respectively, we concluded that the U-rich 3′-overhang contributed to the formation of a more stable crRNA-AsCpf1 complex. From these results, we were able to draw the conclusion that the U-rich 3′-overhang induces more favorable binding between the crRNA and Cpf1, thereby improving the Cpf1 activity.

## Discussion
A CRISPR-based genome-editing platform was engineered by developing altered forms of either effector endonucleases (e.g., Cas9) or gRNAs. Several amino acids involved in interactions with the phosphodiester groups in DNA were mutated to create a CRISPR-Cas9 system with an enhanced specificity and with either mitigated or no off-target incidences[36,37]. Specially designed Cas9 and Cpf1 variants conferred an altered PAM specificity for ver-satile genome editing[19,28]. Cas9 was also engineered to generate effector endonuclease with a nickase activity (nCas9) or nuclease activity-null Cas9 (dCas9) for various purposes[38]. These Cas9 variants were fused to proteins with various functions to enhance the targeting specificity[39] or to modulate gene expressions[40].

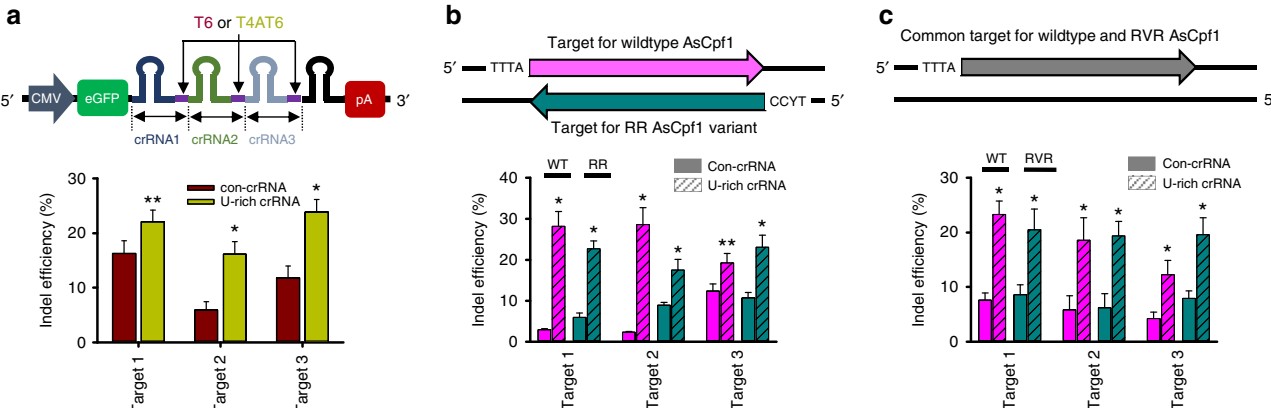

**Fig. 5** Applications of the U-rich crRNA to multiplexed genome editing and PAM-divergent AsCpf1 variants. **a** Simultaneous improvements of the indel efficiencies of multiple targets by an array of U-rich crRNAs. crRNA-encoding sequences with three different targets and one scrambled one were cloned into the 3′-UTR region of the eGFP gene in the pEGFP-C1 vector (Clontech). The U-rich crRNAs have target sequences with a 20-base match plus an additional T4AT6, while each control crRNA only has 23-base match target sequences. HEK-293T cells were transfected with 5 μg each of Cpf1-encoding and crRNA-encoding vectors. The indel efficiency was calculated after normalization with the transfection efficiency as assessed by the green-fluorescent cell counts. **b, c** Application of U-rich crRNA to AsCpf1 PAM variants. *$p < 0.001$, **$p < 0.01$ ($n = 3$), two-tailed Student's $t$ test. **b** Three targets were selected as common targets for the WT and RR variant of AsCpf1, which have TTTA and TYCC PAM sequence in each strand. The indel efficiency of WT AsCpf1 or the RR variant was investigated in the presence of the canonical or U-rich crRNA. *$p < 0.001$, **$p < 0.01$ ($n = 3$), two-tailed Student's $t$ test. **c** Three different targets with a TTTA PAM sequence were subjected to indel mutation by transfecting HEK-293T cells with the WT or the RVR variant of AsCpf1, each of which was guided by either canonical or U-rich crRNA. *$p < 0.001$, ($n = 3$), two-tailed Student's $t$ test

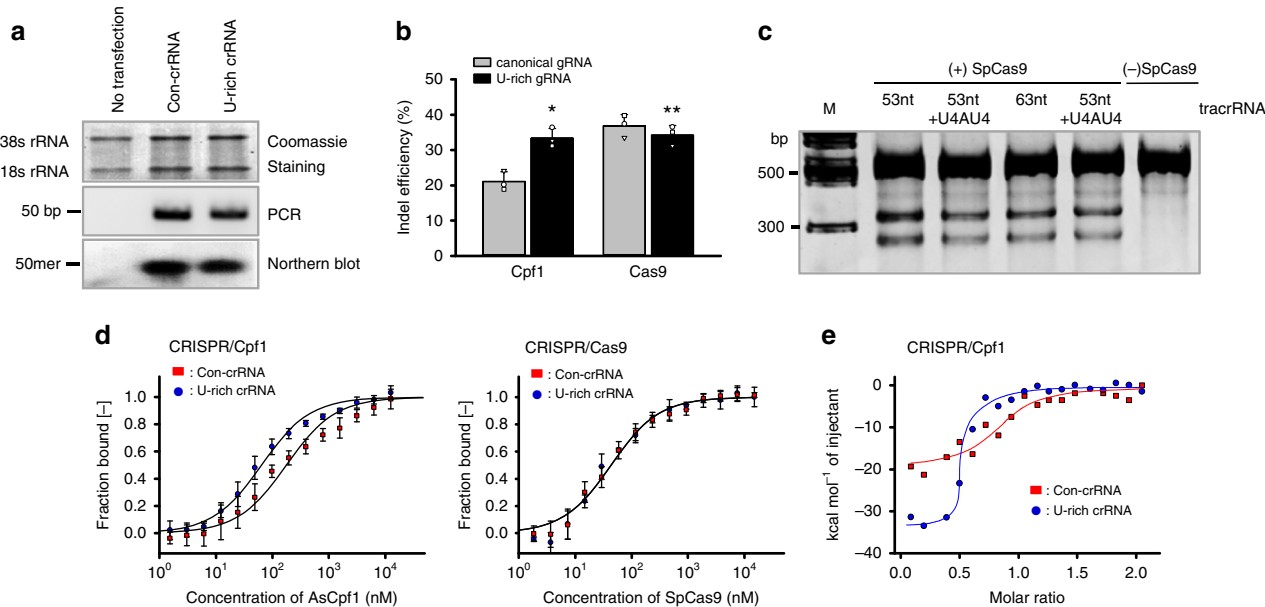

**Fig. 6** Improved binding affinity of the crRNA-AsCpf1 complex by the U-rich 3′-overhang. **a** The endogenous levels of the crRNAs were assessed by northern blot analysis following transfection of the crRNA template PCR amplicons into HEK293-T cells. The intracellular levels of 38s/18s RNA were used as a loading and quality control of the endogenous RNAs. The transfection efficiency was also controlled by the PCR intensities of the crRNA template DNA isolated from the transfected HEK293-T cells. This result is representative of three repeated experiments. **b** The indel efficiency was monitored at a DNMT1 locus following transfection of the AsCpf1 plasmid constructs and synthetic guide RNAs into HEK293-T cells. The 3′-terminal four nucleotides of crRNA and sgRNA for AsCpf1 and SpCas9, respectively, were covalently linked with the phosphorothiate group. *$p = 0.008$, **$p = 0.39$ ($n = 3$), two-tailed Student's $t$ test. **c** The indel efficiency was measured using the crRNA and tracrRNA of the CRISPR-Cas9 system. The truncated crRNAs with a length of 53 nt and 63 nt were compared with those with the U4AU4 3′-overhang. This gel image is representative of three repeated experiments. **d** Binding experiments were conducted using a microscale thermophoresis (MST) system. The binding affinities of guide RNAs and the effector proteins (SpCas9 and AsCpf1) were measured using Monolith NT.115 (NanoTemper Technologies GmbH). All samples were loaded into NanoTemper standard capillaries and repeated three times for each measurement. **e** Isothermal titration calorimetry (ITC) measurements were conducted on an Auto-iTC200 Microcalorimeter (GE Healthcare) at 25 °C in PBS buffer (pH 7.4). The purified recombinant AsCpf1 proteins (5 μM) were titrated with 50 μM synthetic crRNAs with 2 μL injections. $K_a = (1.90 \pm 0.87) \times 10^8$ M$^{-1}$ for the U-rich crRNA and $(1.15 \pm 0.54) \times 10^7$ M$^{-1}$ for the canonical crRNA. $\Delta H = -31.92 \pm 1.79$ kcal/mol and $-22.86 \pm 1.86$ kcal/mol for the U-rich and canonical crRNA, respectively. The values are the average of three independent experiments

Alternatively, gRNA was designed to enable genome editing in a more reliable manner. tracrRNA and crRNA were fused into an sgRNA for convenient genome editing without harming genome-editing properties[41]. Fu et al.[22] suggested that a truncated sgRNA improved the target specificity without compromising the indel efficiency when the seed region was conserved. By integrating these modifications, it became possible to perform genome-editing using CRISPR-Cas9 with improved specificities and conveniences. To our knowledge, however, there have been no studies in which the engineering efforts have led to a significant improvement of indel efficiency. We were able to improve the indel efficiency of Cpf1 by up to approximately 13-fold with a 3′-uridinylate-rich crRNA, and this improved Cpf1 activity was comparable to or even higher than SpCas9 activity. Furthermore, the engineered crRNA did not affect the target specificity of Cpf1.

Several studies have tried to apply a genome-editing platform to the development of gene therapy. These studies were done by applying a first-generation genome-editing technology, zinc-finger nuclease, to knockout the CCR5 gene in T cells to treat HIV infection[42], which showed promising results in past clinical trials[43]. In addition, clinical trials adopting in vivo gene therapy approaches are either on-going or are currently expanded[11]. Considering the efficiency and robustness of the CRISPR technology, it is expected that efforts to develop genome-editing-based gene therapy will drastically increase in preclinical and clinical settings. As a therapeutic agent, the CRISPR-Cpf1 platform has several advantages over CRISPR-Cas9. The generally smaller sizes of the Cpf1 gene compared with that of Cas9 provides more versatile opportunities particularly when delivered using AAV-based viral vectors. crRNA, which is half the size of the sgRNA for Cas9, provides additional advantages considering the increased error rates in the synthesis of longer nucleotides. Furthermore, the low risk of off-targeting is an important property of Cpf1 compared with Cas9[19–21]. We assume that these merits have been compromised by the target-dependent, insufficient genome-editing efficiency of Cpf1. Given the innate merits of Cpf1 over Cas9, the improved activity would be a critical determinant for Cpf1 to be used as a more versatile genome-editing tool.

Despite efforts to elucidate the mechanism underlying the improvement of the Cpf1 performances by the U-rich crRNA, we could not reach a plausible conclusion to explain why the occupation of uridinylates at the 3′-end region of the crRNA enhances the DNA double-strand breakage activity of Cpf1. This failure stemmed, at least partly, from the status quo of the incomplete structural elucidations based on the co-crystallized complex consisting of Cpf1, crRNA, and target dsDNA, as well as the incomplete understanding of how programmable nucleases recognize and cleave endogenous DNA inside eukaryotic cells. Although mechanistic elucidations remain elusive, it is interpreted that the improved binding affinity by the 3′-overhang contributes to the increased binding of the crRNA and Cpf1 protein inside cells and possibly to the stabilization of the formation of the ribonucleoprotein complexes. Full elucidations of the relevant processes might provide a further opportunity to enhance the efficiency of Cpf1 in vivo.

## Methods

**Cell culture and transfection**. HEK-293T cells (293T/17, ATCC) were maintained in high-glucose Dulbecco's modified Eagle's medium (DMEM), supplemented with 10% heat-inactivated fetal bovine serum (Corning) and 1 mM penicillin/strepto-mycin at 37 °C in an incubator with 5% $CO_2$. Cell transfection was performed through either electroporation or lipofection methods. For electroporation, 2–5 µg of AsCpf1-encoding, LbCpf1-encoding, or SpCas9-encoding plasmid vectors (Addgene) were transfected into $5 \times 10^5$–$1 \times 10^6$ HEK-293T cells with 1–3 µg crRNA-encoding or sgRNA-encoding PCR amplicons using a Neon electroporator (Invitrogen). When necessary, chemically synthesized crRNAs (Bioneer) were used

for transfection instead of PCR amplicons. For lipofection, 3–15 µL FuGene reagents (Promega) was mixed with 1–5 µg of AsCpf1-encoding, LbCpf1-encoding, or SpCas9-encoding plasmid vectors plus 3–15 µg of PCR amplicons for 15 min. The mixture (300 µL) was added to 1 mL DMEM media in which $5 \times 10^5$ cells were plated 1 day before transfection, and cells were grown in the presence of the mixture for 48 h. After incubation, cells were harvested, and genomic DNA was prepared either manually using a PureHelix™ Genomic DNA Preparation Kit (NanoHelix) or using a Maxwell™ RSC nucleic acid isolation workstation (Promega). pSpCas9(BB)-2A-GFP (PX458), pY010(pcDNA3.1-hAsCpf1), and pY016 (pcDNA3.1-hLbCpf1) were gifts from Feng Zhang (Addgene plasmid #48138, #69982, #69988, respectively). Target information throughout the present study was compiled in Supplementary Table 4.

**AsCpf1 PAM variants**. Site-directed mutagenesis was performed on a Veriti Thermal Cycler (Life Technologies) using pY010 plasmid vector as a template and mutagenic primers. S542R mutation was created using a mutagenic primer pairs (5′-TACACTGGCCAGAGGCTGGGACG-3′/5′-CGTCCCAGCCTCTGGC-CAGTGTA-3′). K607R and K548V/N552R mutations were separately created using additional mutagenic primers (5′-GATGATCCCAAGGTGCAGCACCC-3′/5′-GGGTGCTGCACCTTGGGATCATC-3′ and 5′-GTGGAGAAGAACA-GAGGCGCCATCCTGTTT-3′/5′-TCTGTTCTTCTCCA-CATTCACGTCCCAGCC-3′, respectively). Briefly, an initial denaturation (3 min, 94 °C) and 25 cycles of denaturation (20 s, 95 °C), annealing (40 s, 62 °C), and extension (10 min, 72 °C) was performed in the presence of 100 ng of plasmid templates and 15 pmol each of mutagenic primers in 50 µL of Toyobo KOD mixture (Takara). Ten microliters of the PCR products were incubated with 2 µL of DpnI (New England Biolabs) at 37 °C for 2 h. After heat denatured at 62 °C for 20 min, the incubated mixture (5 µL) was used to transform BL21(DE3) E. coli cells. Mutagenesis was confirmed by Sanger sequencing analysis.

**Unbiased in vitro experiment**. crRNA library oligonucleotides with a randomized 11-nt sequence at the 3′-end were synthesized and quality controlled (Integrated DNA Technologies) so that each crRNA occupies the equal molar ratio. The oligonucleotide library was cloned into a pET21 plasmid vector using the sequence-independent and ligation-independent cloning method[44]. The cloned plasmid constructs were used to transform BL21 (DE3) E. coli cells and secured a colony-forming unit of ≥$10^8$ CFU/mL. The CFU value was calculated by counting the colonies of serially diluted transformed cells on the ampicillin (+) plates. The transformed cells were grown in LB broth supplemented with 50 ng/mL ampicillin until the optical density reaches 0.6. The competent cells ($2 \times 10^{10}$ cells/mL) were transformed with either dCpf1 or Cpf1-carrying pET-28a(+) plasmid vector (50–200 ng) using a Gene Pulser Xcell electroporator (Bio-Rad). The transformed cells were plated onto agar plates supplemented with ampicillin and kanamycin plus 0.1 M isopropyl β-D-1-thiogalactopyranoside. Colonies formed onto each plate were pooled, from which plasmid vectors were purified. The plasmid vectors were subjected to deep sequencing analysis to calculate the frequency of A/T/G/C at each position of crRNA using an Illumina HiSeq X Ten Sequencer at Macrogen (South Korea).

**Purification of recombinant AsCpf1**. A codon-humanized cpf1 gene from Acid-aminococcus sp. was cloned into a pET-28a(+) plasmid vector (Invitrogen), and the vector construct was used to transform BL21(DE3) E. coli cells. An E. coli transformant colony was grown at 37 °C in LB broth in the presence of 50 nM kanamycin until reaching an optical density of ca. 0.7. Subsequently, cells were incubated at 30 °C overnight in the presence of 0.1 mM isopropylthio-β-D-galac-toside to induce the production of the recombinant proteins. Cells were subsequently collected by centrifugation at $3500 \times g$ for 30 min and subjected to disruption by sonication. Cell lysates were cleared by centrifugation at $15,000 \times g$ for 30 min and subsequently by filtration through a 0.45-µm syringe filter (Millipore). The cleared lysates were loaded onto a $Ni^{2+}$-affinity column using an FPLC Purification System (ÄKTA Purifier, GE Healthcare). Alternatively, the recombinant AsCpf1 was purified in an Automated Protein Production System (ExiProgen, Bioneer) by adding 1 µg of the gene construct into in vitro translation mixtures. The concentration of the produced proteins was electropherometrically determined on a Coomassie blue-stained sodium dodecyl sulfate-polyacrylamide gel electrophoresis (SDS-PAGE) gel using bovine serum albumin (BSA) as a standard.

**In vitro DNA cleavage by AsCpf1**. PCR amplicons carrying DNA sequence of 5′-CTGATGGTCCATGTCTGTTACTC-3′ following a TTTC PAM was cloned into a T-Blunt vector (Solgent). The vector constructs were amplified in DH-5α E. coli cells and purified using a HiGene™ DNA Purification Kit (Biofact). The target vector (20 ng/µL) was incubated with purified recombinant AsCpf1 proteins (50 ng/µL) and chemically synthesized crRNAs (10 ng/µL) at 37 °C for 30–60 min. The incubated mixtures were either resolved on 10% SDS-PAGE gels for quantification of cleaved products or, alternatively, used to transform DH-5α E. coli-competent cells by imposing heat shock at 42 °C for 2 min. The transformed cells were plated onto LB agar plates containing ampicillin (50 ng/µL) and grown in an incubator at 37 °C. The number of colony formed was counted to derive crRNA-dependent DNA cleavage efficiency of AsCpf1.

**Indel quantification**. T7 endonuclease I (T7E1) assays were performed to assess indel efficiency by AsCpf1, LbCpf1, or SpCas9 at targeted loci in HEK-293T cells. PCR products were obtained by PCR amplifications of targeted loci using a Solg™ Pfu-based PCR Amplification Kit (SolGent). PCR products (100–300 μg) were incubated with 10 U of T7E1 enzyme (New England Biolabs) in 25 μl of reaction mixture at 37 °C for 1 h. Twenty microliters of reaction mixtures were directly loaded onto 10% SDS-PAGE gels, and digested products were resolved run in a Tris/Borate/EDTA buffer system. After staining in ethidium bromide solutions, gels were digitalized in a Printgraph 2M Gel Imaging System (Atto). Digitalized images were analyzed to calculate indel efficiency using the Image J software.

**Assessment of off-target activity**. For biased off-target analysis, potential off-target sites were selected using Cas-OFFinder (23; http://www.rgenome.net/cas-offinder; Supplementary Table 2) with a criterion of less than two bulges and mismatches. After transfection with AsCpf1 vector constructs and crRNA-encoding PCR amplicons, HEK-293T cells were grown in DMEM for 2 days. The on-target and potential off-target sites were amplified by nested PCR and used for library construction. Each library was purified using Agencourt AMPure XP (Beckman Coulter) and quantified by the Picogreen method using the Quanti-iT PicoGreen dsDNA Assay Kit (Invitrogen). After the library size was confirmed using the Agilent 2100 Bioanalyzer System (Agilent technologies), quantitative PCR analysis was conducted to match the input amount and the appropriate clusters as proposed by Illumina. Subsequently, paired-end sequencing was conducted using MiSeq Reagent Kit V3 (Life Sciences) by the Illumina MiSeq sequence platform. Primer sequences were removed from each raw data using the Cutadapt tool (version 1.14). The trimmed sequences were clustered and were subjected to sequence comparisons. Indel mutations observed within 23-nt target sequences were considered as genetic alterations by off-target activity. Alternatively, a DNMT1 target site of HEK-293T cell line was PCR amplified following inducing indel mutations by electroporation of 5 μg of AsCpf1 vector constructs and 3 μg of crRNAs with an on-target or one-base mismatch sequences into $2 \times 10^6$ HEK-293T cells. The indel efficiency was measured on SDS-PAGE gels through T7E1 digestion assay. Unbiased, genome-wide off-target analysis was also performed as described previously[20]. Briefly, genomic DNA was isolated from HEK-293T cells using using a DNeasy Tissue Kit (Qiagen). Ribonucleoprotein complex was formed by pre-incubating AsCpf1 protein and crRNA at 300 and 900 nM, respectively, at room temperature for 10 min. Genomic DNA (8 μg) was incubated with RNP complexes in a reaction buffer comprising 100 mM NaCl, 10 mM $MgCl_2$, 100 μg/mL BSA, and 50 mM Tris-HCl (pH 7.9) in a reaction volume of 400 μl at 37 °C for 8 h. Digested genomic DNA was purified using a DNeasy Tissue Kit (Qiagen) following treatment with RNase A (50 μg/mL) to degrade crRNAs. Cpf1-digested genomic DNA in the presence of either canonical or U-rich crRNA was subjected to whole-genome sequencing (WGS) at a sequencing depth of 30× to 40× using an Illumina HiSeq X Ten Sequencer. A DNA cleavage score was assigned to each nucleotide position across the entire genome, using WGS data, according to the equation as mentioned previously[20]. These equations assume that Cpf1 produces 5′ 1-nt to 5-nt overhangs independent of the identity of crRNA. DNA cleavage scores above the cutoff value of 2.5 with six or fewer mismatches from the on-target sequence were computationally identified via Digenome-seq program (https://github.com/chizksh/digenome-toolkit2).

**Binding experiments**. Binding experiments were conducted using an ITC and MST. First, ITC measurements were conducted on an Auto-iTC200 Micro-calorimeter (GE Healthcare) as reported elsewhere[26]. Briefly, chemically synthesized canonical or U-rich crRNA (50 μM) was titrated into a titration cell at 2 μL per injection containing 5 μM purified recombinant AsCpf1 protein at 25 °C in PBS buffer (pH 7.4). The data analyses were carried out using the MicroCal OriginTM software (GE Healthcare). The calculated values are the average of three independent experiments. The binding affinities of the gRNAs and the effector proteins (SpCas9 and AsCpf1) were measured using Monolith NT.115 (NanoTemper Technologies GmbH). Chemically synthesized crRNAs (IDT Technologies) were labeled with Cy5 fluorescent dye. Purified recombinant AsCpf1 with varying concentration (0.25 nM–50 μM) was mixed with 8 nM of labeled RNA in PBS buffer containing 0.05% Tween-20 and 0.05% BSA. Measurements were performed at 24 °C using 5% light-emitting diode (LED) power and 20% MST power. For the Cas9 MST experiments, the Cy5-labeled crRNAs were hybridized to the tracrRNAs at an equimolar ratio. Briefly, two RNA oligos resuspended in Nuclease-Free Duplex Buffer (IDT Technologies) were heated at 95 °C for 5 min and allowed to cool to room temperature. Purified SpCas9 protein with varying concentration (0.1 nM–15 μM) was mixed with 8 nM labeled RNA in 20 mM HEPES (4-(2-hydroxyethyl)-1-piperazineethanesulfonic acid) buffer (pH 7.4) containing 150 mM KCl, 0.05% Tween-20, and 0.05% BSA. Measurements were performed at 24 °C using 20% LED power and 20% MST power. All samples were loaded into NanoTemper standard capillaries and repeated at least three times for each measurement. The binding affinity data were analyzed using the NanoTemper analysis software.

**Northern blot analysis**. Total RNA was extracted from HEK-293T cells using the Maxwell RSC miRNA Tissue Kit (Promega), following the manufacturer's instructions. For each sample, 0.3–0.5 μg of isolated RNA were separated on a 1% agarose/16% formaldehyde gel after denaturation for 15 min at 65 °C in RNA denaturation buffer (20% formaldehyde, 50% formamide, 50 mM MOPS (3-(N-morpholino)propanesulfonic acid), pH 7.0). Afterwards, RNA was transferred overnight to a positively charged nylon membrane by capillary transfer in 10× SSC. After crosslinking, RNA was pre-hybridized at 50 °C for 30 min in a pre-warmed Digoxigenin (DIG) Easy Hyb (Roche) and hybridized overnight at 50 °C with 20–50 ng/mL PCR DIG probe, generated with PCR DIG Labeling Mix (Roche) and denatured at 96 °C for 5 min. Blots were washed and immunodetected with the anti-DIG-AP Fab fragments (Roche). The target RNA–DNA probe hybrids were visualized by chemiluminescent assay using the CDP-Star substrate (Roche). The probe sequences are listed in Supplementary Table 5.

**Statistical analysis**. Statistical tests of indel efficiency were performed in Sigma Plot using a two-tailed Student's $t$ test. $P$ values <0.05 were considered significant and are presented in the Legends section.

**Uncropped images**. Uncropped images of the blots in Fig. 6 can be found in Supplementary Figure 6.

## Data availability

The deep sequencing data were deposited at the NCBI Sequence Read Archive under accession number SRP115978. All other data that support the findings of the present study are available from the corresponding author upon request.

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

## Acknowledgements

This work was supported by the "Bio & Medical Technology Development Program" through the National Research Foundation funded by the Ministry of Science and ICT (NRF-2016M3A9B6903343), and "R&D Convergence Program" of National Research Council of Science and Technology (CAP-15–03-KRIBB).

## Author contributions

Y.-S.K. and J.-H.K conceived the study and designed the experiments. S.B.M. and J.M. L. performed the overall experiments. J.G.K., N.-E.L., D.-I.H., D.Y.K., S.H.K., and K.Y. performed the large-scale validation study. D.K. performed the genome-wide off-target analysis. Y.-S.K. wrote the manuscript. All authors approved the manuscript.

## Additional information

**Competing interests:** The authors declare no competing interests.

