## [Peer Review File · Nature Communications]

Reviewers' comments:

Reviewer #1 (Remarks to the Author):

The authors have addressed part of the previous comments. However, there are a substantial set of concerns that still remain:

1. The effect of oligo-U overhang remain minimal to various other types, such as A8, C8 or mixture of A/U/C oligos.
2. More seriously, in the figure in response letter, out of the five loci tested, i.e. LINC01551 P2RX5- TAX1BP3, TYW1B, Intergene-1 and Intergene-2, 4/5 (80%) of the cases showed that the con-Cpf1 and U-Cpf1 showed almost identical cutting efficiency, thus might directly negate the conclusion of the paper.
3. Some of the data in the main seem to be selection of the best cases, which are contradictory to the supplemental data.

thus, unless a fully unbiased, compelling evidence were shown, it remains uncertain that the U-overhang substantially improve Cpf1-editing.

Reviewer #2 (Remarks to the Author):

The authors addressed all the issues I have raised.

Answers to the reviewer's comment

1. The effect of oligo-U overhang remain minimal to various other types, such as A8, C8 or mixture of A/U/C oligos.

Fig. 1g (in the revised manuscript) indicates that the U-rich overhang significantly enhance the cleavage of target DNA, in which p -values were all less than 0.05, compared to other types. The difference between A8, C8, and the mixture of A/U/C is minimal, but the oligo-U exerted significantly improved effects. The marked difference can also be observed in other assays including Fig. 1b and 2b.

2. More seriously, in the figure in response letter, out of the five loci tested, i.e. LINC01551, P2RX5-TAX1BP3, TYW1B, Intergene-1 and Intergene-2, 4/5 (80%) of the cases showed that the con-Cpf1 and U-Cpf1 showed almost identical cutting efficiency, thus might directly negate the conclusion of the paper.

Our large-scale data indicated that the U-rich overhang improved the Cpf1 activity with a different extent. Some of targets were dramatically improved while others were not. We don't know how this target-to-target difference occurs, but it is true that most of the targets are improved with different extents. Please refer to the Supplementary Table 1. Among the five targets you mentioned, LINC01551 showed no noticeable improvement, but there were, albeit marginal, more or less improvement for the last 4 targets. As I noted in the previous response letter, the effects of U-rich overhang are more pronounced in the targets with low indel efficiency, for example, P2RX5-TAX1BP3 in this case. One more thing I'd like to mention is that the five targets are in fact included in the large-scale 115 targets, and the statistical analysis was implemented with the five targets included. It is possible that a restricted, small sampling may negate our conclusion, but please understand that our conclusion was drawn based on the large-scale analysis.

3. Some of the data in the main seem to be selection of the best cases, which are contradictory to the supplemental data. Thus, unless a fully unbiased, compelling evidence were shown, it remains uncertain that the U-overhang substantially improve Cpf1-editing.

Many of data in the main text were derived from the DNMT1 target because DNMT1 is one of the targets frequently shown in the major genome-editing papers. Please see to it that we have much more targets showing more dramatic effects in the supplementary data, compared to DNMT1. Despite all these rebuttals, however, **we agree with your claim that**

our conclusion be derived from a fully unbiased assay because we did not test every possible combination of A/T/G/C-tail.

For this, we designed the following experiment:

Firstly, we prepared crRNAs with a library of 3-tails. crRNA library oligonucleotides with a 3'-end sequence library (4^{11}) were synthesized and quality-controlled so that each crRNA occupies the equal molar ratio. Each crRNA was designed to have a 17-nt on-target sequence and 11-nt (N_{11}) random sequences as below. We thought that the shortened on-target sequence (here, 17-nt, not 20- or 23-nt) can help define the essential on-target length for Cpf1 activity as well as additional regulatory sequence.

• **Target sequence (Amp^R gene)**

5'-TTTTCAGGGTTATTGTCTCATGAGCGgatacatattt-3'
 PAM 23-nt target

• **crRNA library**

The oligonucleotide library was cloned into a pET21 plasmid vector using the sequence- and ligation-independent cloning (SLIC) method (Li & Elledge, Methods Mol Biol, 2012). The cloned plasmid vector was used to transform BL21 (DE3) *E. coli* cells and secured a colony forming unit of $\geq 10^8$ CFU/mL ($4^{11} = 4.2 \times 10^6$). The CFU value was calculated by counting the colonies of serially diluted transformed cells on the ampicillin (+) plates (1.56×10^8 CFU/mL).

Dilution factor	Number of colony
10^5	1,676
10^6	144
10^7	15
Colony forming unit= 1.1558×10^8 CFU/mL	

The transformed cells were grown to make electro-competent cells carrying the crRNA-encoding plasmid library. The competent cells (2×10^{10} cells/mL) were transformed with either dCpf1 or Cpf1-carrying pET-28a(+) plasmid vector (50-200 ng). The use of dCpf1 vector was to normalize the content of each crRNA in competent cells prior to the transformation using the Cpf1 vector. The transformed cells were plated onto agar plates supplemented with ampicillin and kanamycin plus 0.1 M IPTG. Colonies formed onto each

plate were pooled, from which plasmid vectors were purified. The plasmid vectors were subjected to deep sequencing to calculate the frequency of A/T/G/C at each position of crRNA. The basic rationale for our strategy is that a 'good crRNA' cleaves the ampicillin-resistant pET21 vector more efficiently, which, in turn, makes the *E. coli* cells carrying the 'good crRNA' less prone to survive in the amp(+) plates. This negative selection method was established previously (Kleinstiver et al., Nature Biotechnology, 2015) and also used in our experiments in this study with necessary modifications (Supplementary Figure 2&3).

The deep sequencing data revealed that the crRNA-encoding plasmid DNA library was made so that A/T/G/T occupied almost equal molar ratio at each position as assessed by dCpf1 treatment. Marginal variations were normalized by the values obtained by dCpf1 treatment. In contrast, there were significant differences in the frequency of each nucleotide in a position-dependent manner when AsCpf1 was treated (Fig. 1c).

From the deep sequencing data, we derived the probability values that point to the positional optimal crRNA sequence as follows (Fig. 1d).

From the result, we were able to extract several important clues as follows:

- 1) AsCpf1 shows a higher cutting efficiency when the target length is set to be 20-nt, not 23-nt. Please refer to the figure above to see that the efficient crRNA sequence matches the on-target sequence up to 20-nt (...TGA).
- 2) The sequence of three nucleotides (that is, position 21-23) did not match the on-target sequence. Noticeably, G at the position 21 showed the lowest probability. Thus, the corresponding sequence should be replaced with UUU when crRNA is designed.
- 3) The U-effect was extended to the position 27 but become decreased as the U-tail becomes longer. There was no significant difference among A/T/G/C at the position 28.
- 4) When crRNA is transcriptionally produced from the DNA template, 5x or more Ts act as a termination signal and thus be replaced with non-T nucleotide at the position next to 4xT. Because the second highest efficiency was obtained by A in the figure above (although there are significant difference among A/G/C), the sequence at the position (that is, the position of 25th from the PAM) should be set to A, which is in line with our previous result (Fig. 2e).

From these results, we suggest the optimized conformation of crRNA for efficient genome editing using Cpf1 as follows:

- a) When chemically synthesized, crRNA with 20-nt on-target sequence plus U8 renders the highest cutting efficiency.
- b) When transcriptionally produced, crRNA-producing DNA template should contain 20-nt on-target sequence plus T₄A_T6. This make-up generates the U-rich (U₄AU₄) 3'-overhang.

All these corrections were made in the main text (p.5), Fig. 1c-d, Supplementary Fig. 2, Figure legend, and Methods.

When we received your comments in the previous manuscript, we thought that your claims

were too harsh, and that the previous results we have presented may be sufficient to generalize our claim. But, after we obtained this unbiased data in compliance with your suggestion, we found that the experiment was essential to back up our conclusion and, more importantly, we were able to be more convinced of our results. Thus, we deeply appreciate your shrewd pointing-out and evaluate that your comments provided an important opportunity to make our paper more technically sound one. Although we didn't elucidate the molecular mechanism underlying the U-rich tail effect yet, we are attempting to elucidate by adopting the single-molecule FRET method. Your comments were tremendously helpful for the related works.